# Increased homicide played a key role in driving Black-White disparities in life expectancy among men during the COVID-19 pandemic

Michael T. Light ●*, Karl Vachuska

Department of Sociology, University of Wisconsin-Madison, Madison, Wisconsin, United States of America

* mlight@ssc.wisc.edu

**Data Availability Statement:** All data needed to evaluate the conclusions in the paper are present in the paper and/or the Supplementary Materials.

## Abstract

Disparities in life expectancy between Black and White Americans increased substantially during the COVID-19 pandemic. During the same period, the US experienced the largest increase in homicide on record. Yet, little research has examined the contribution of homicide to Black-White disparities in longevity in recent years. Using mortality data and population estimates, we conduct a comprehensive decomposition of the drivers of Black-White inequality in life expectancy and lifespan variability between 2019 and 2021 among men. We find that homicide is one of the principal reasons why lifespans have become shorter for Black men than White men in recent years. In 2020 and 2021, homicide was the leading contributor to inequality in both life expectancy and lifespan variability between Black and White men, accounting for far more of the racial gap in longevity and variability than deaths from COVID-19. Addressing homicides should be at the forefront of any public health discussion aimed at promoting racial health equity.

## Introduction

The COVID-19 pandemic precipitated a staggering drop in U.S. life expectancy and substantially widened Black-White disparities in longevity [1,2]. Not only did the disease burden of the COVID-19 virus fall disproportionately on Black Americans, but the cascading disruptions during the pandemic also took a heavy toll on the Black community [3], including homicide. Following notable declines in homicide since the early 1990s, the U.S. homicide rate rose 30% between 2019 and 2020 –the largest one-year increase in over a century [4]. Homicide went up again in 2021, to the highest point in more than two decades [5], and Black men bore the brunt of these stark homicide increases. Between 2019 and 2020, the non-Hispanic Black male homicide rate increased 39%, from 43.8 to 61.0 (per 100,000 Black men in the population). For non-Hispanic White men, the increase was 22% (compare 3.6 to 4.4 per 100,000 White men). As a result, the Black-White homicide ratio among men shot up from 12:1 in the year immediately preceding the pandemic, to approximately 14:1 in 2020 and 2021.

Computer code used to produce the study results will be made available on OpenICPSR. The mortality data can be accessed from the NVSS at https://www.cdc.gov/nchs/data_access/vitalstatsonline.htm#Mortality_Multiple and the population estimates are available from SEER at https://seer.cancer.gov/popdata/download.html.

**Funding:** This research is supported by the Romnes Faculty Fellowship provided by the University of Wisconsin-Madison Office of the Vice Chancellor for Research and Graduate Education with funding from the Wisconsin Alumni Research Foundation. The funders had no role in study design, data collection and analysis, decision to publish, or preparation of the manuscript.

**Competing interests:** The authors have declared that no competing interests exist.

Yet, despite these trends, there has been limited research on the contribution of homicide to Black-White disparities in life expectancy during the pandemic. This is a notable gap given that homicide was a major contributor to racial inequality in life expectancy even before the pandemic [6]. Moreover, homicide has an outsized impact on longevity because, unlike COVID-19, it disproportionately afflicts the young (especially young men), among whom homicides have become disquietingly common [7]. Indeed, gun deaths increased 50% among U.S. children under 18 between 2019 and 2021, the majority of which were homicides [8].

Against this backdrop, in this study we provide a detailed decomposition of the specific causes of death that drove the changes in Black-White (B-W) life expectancy in 2020 and 2021 (relative to 2019), with an emphasis on the relative import of homicide to these mortality shifts. Because men suffer disproportionally from homicide, we focus on male mortality throughout our analysis. We also decompose the sources of *lifespan variability* (the variance in age at death) between Blacks and Whites during this period. Although life expectancy is an essential indicator of population health, it also masks important heterogeneity because two populations with similar mean ages at death can exhibit notably different dispersion around the mean [9]. In this regard, greater lifespan variability is a fundamental inequality because it translates to greater uncertainty about age at death, which can prompt individuals to discount their future [10]. This is particularly consequential in the context of pervasive violence, as a sense of "futurelessness" can contribute to yet further crime and violence [11]. Our complementary analyses of both life expectancy and lifespan variability thus provide a comprehensive look at the link between homicide and the changes in mortality inequality between Black and White men in recent years.

Our focus on both 2020 and 2021 is important because previous decompositions of racial disparities in life expectancy during the pandemic concentrated mainly on 2020 [2]. However, mortality dynamics have changed appreciably since the initial onset of COVID-19. Notably, 2021 saw the widespread dissemination of multiple COVID-19 vaccines and the narrowing and then vanishing of the Black-White gap in COVID-19 deaths [12]. And yet, the racial gap in life expectancy declined only slightly in 2021 compared to 2020 [3]. Thus, an updated understanding of the drivers of Black-White inequality in life expectancy and variability is crucial for informing public health conversations about ameliorating racial disparities in mortality.

## Materials and methods

### Data and life table construction

Mortality information comes from the publicly available US multiple cause of death data files, from the National Vital Statistics System division of the National Center for Health Statistics (https://www.cdc.gov/nchs/data_access/vitalstatsonline.htm#Mortality_Multiple). For 2019–2021, we aggregated mortality statistics by cause of death, 1-year age category (up to age 100), sex, and race/ethnicity, where causes of death are coded according to ICD-10 codes [13]. Drawing from prior research [9], we collapse the ICD-10 codes into 20 causes (see S3 Table for the ICD-10 codes for each cause). Population estimates by age category, gender, and race/ethnicity come from the yearly population estimates published by the Surveillance, Epidemiology, and End Results Program (https://seer.cancer.gov/popdata/download.html). For 2021, single-year population estimates for the 85+ age groups are calculated by projecting forward the 2020 population one year using 2020 mortality rates. These projections are rescaled to have the same population size as SEER's 2021 estimate for the 85+ age group by race and sex.

With these data, we constructed multiple-decrement life tables by cause of death, age, sex, and race for the period 2019–2021 using standard demographic techniques in public health

and demography [2]. We estimate life expectancy as the average number of years a synthetic cohort of newborns is expected to live if they were to experience the mortality rates observed in a given year (details for our procedures are shown in the *Supporting Information*). This methodology is apt for comparing different groups over time because it is not sensitive to differences in population size or age structure.

## Decomposition of life expectancy and lifespan variability

We decompose inequalities in life expectancy by cause with the following equation:

$$_n\Delta_x^i = {_n}\Delta_x * \left( \frac{_nm_x^i(2) - {_n}m_x^i(1)}{_nm_x(2) - {_n}m_x(1)} \right)$$

Where $_n\Delta_x^i$ represents the difference in life expectancies between groups 1 and 2, attributed to cause $i$ in the age-category beginning with age $x$ and going through age $x+n$, $_nm_x^i(2)$ represents the age and cause-specific mortality rate for cause $i$ between ages $x$ and $x+n$ for group 2, $_nm_x(2)$ represents the age-specific mortality rate between ages $x$ and $x+n$ for group 2. This equation separates the total contribution of a given age group into the different causes of death within that age category. Consequently, the total contribution for any given cause of death is the sum of the contributions for that cause across age categories.

To estimate which causes of death contributed to changes in inequality over time, we extend our analysis to four groups: non-Hispanic Blacks (1) and non-Hispanic Whites (2) in 2019, and non-Hispanic Blacks (3) and non-Hispanic Whites (4) in 2020 (and 2021). To capture changes in inequality in life expectancy between these groups, we calculate the difference in $_n\Delta_x^i$ at two points in time. To determine the proportion of change in inequality attributed to each cause of death, we divide the differences in $_n\Delta_x^i$ by the total change in life expectancy inequality.

Lifespan variability–the variance in age at death–is also based on the life tables, where the age of death for each age category is calculated as the mid-point of the age interval. For the 100 + age category, we assume the age of death is 102.5. Life expectancy variance is calculated from the predicted number of deaths in each age category using the following formula:

$$S^2 = \frac{\sum \left( _nd_x * y_x^{x+n} - \bar{y} \right)^2}{\sum _nd_x}$$

where $\bar{y} = \frac{\sum _nd_x * y_x^{x+n}}{\sum _nd_x}$ and $y_x^{x+n} = x + n/2$ for all age intervals except for the open-ended interval, where it is equal to 102.5. This calculation produces the squared standard deviation for life expectancy for each age group. The variance attributable to cause $i$ is calculated as:

$$S_i^2 = \frac{\sum \left( _nd_x^i * y_x^{x+n} - \bar{y} \right)^2}{\sum _nd_x}$$

This decomposition yields the number of squared years that can be attributed to each cause of death. The percentage of life expectancy variance that can be attributed to a specific cause $i$, is equivalent to the difference in $S_i^2$ between the two groups divided by the difference in $S^2$ between the two groups. This method is equivalent to the Nau and Firebaugh [14] approach for estimating "gross cause-specific contributions" to differences in life expectancy variance.

To capture changes in life expectancy variance inequality, we calculate the difference in differences in $S_i^2$ between groups at two points in time. To determine the proportion of changes

in inequality caused by each cause of death, we divide the difference in differences in $S_i^2$ between groups by the total change in the inequality of life expectancy variance.

## Results

### Life expectancy

Whereas Black men were expected to live 71.4 years in 2019, this dropped to 67.7 years in 2020. For White men, the corresponding decline was only from 76.4 years to 74.9 years. As a result, the racial gap jumped from 5.0 to 7.2 years in lower life expectancy for Black men relative to White men. These findings align with those reported in prior research [2]. To put this in perspective, the disparity in 2020 was greater than the B-W life expectancy gap observed for males in 2000 (6.6 years) [15], and nearly equaled the gap observed in 1990 (8.2 years) [16]. Stated differently, the first year of the pandemic erased over two decades of progress in reducing inequality in longevity between Black and White men.

Fig 1 decomposes the contributions of specific causes of death to these marked changes in racial inequality in life expectancy between 2019 and 2020, 2019 and 2020, and 2020 and 2021. For parsimony, the figures throughout this article display the top five drivers of Black-White mortality disparities in 2019 (plus COVID-19), and the full results for all 20 causes of death are shown in the Supporting Information.

We begin with the 2019–2020 comparison. Unsurprisingly, deaths from COVID-19 were the largest contributing factor, at 60%. But this means that much of the change in B-W

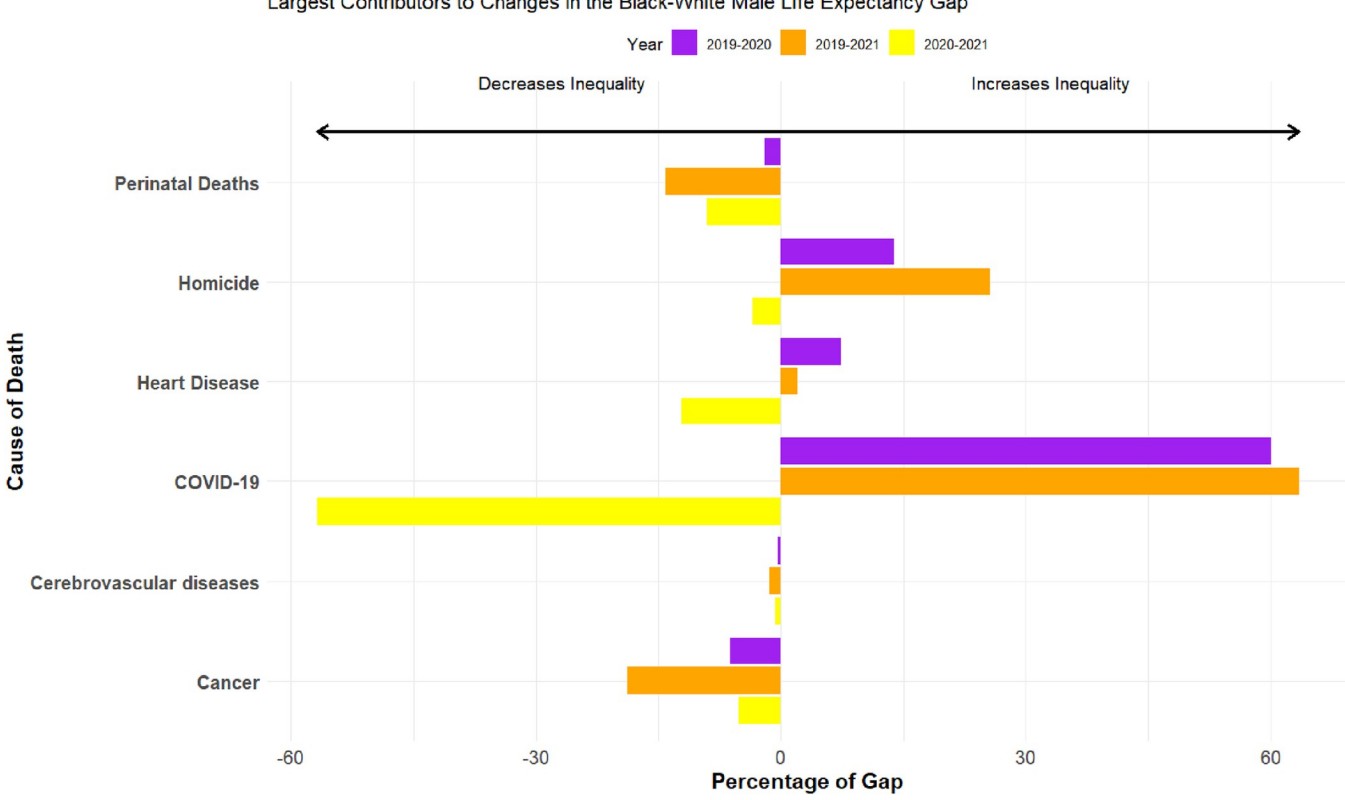

**Fig 1. Cause-specific components of the *changes* in Black-White life expectancy in the United States from 2019–2021 among Males.** *Notes*: For White men, life expectancy in 2019, 2020, and 2021 was 76.4, 74.9, and 74.2 years, respectively. For Black men, life expectancy was 71.4, 67.7, and 68.1 years in 2019, 2020, and 2021.

inequality in life expectancy is attributable to shifts in other causes of mortality in the pandemic's first year. Most notable were homicides at 14%, followed by heart diseases at 7% (accidental poisonings also accounted for approximately 7% of the change in B-W life expectancy inequality, see Supporting Information). Although homicides were not among the top-10 leading causes of death in 2020 and only accounted for 2.9% of the decline in life expectancy between 2019 and 2020 for the overall population [17], changes in homicide mortality were the second leading factor driving the growth in B-W inequality in life expectancy among men. It is important to place this result in recent historical context. In the decades leading up to the pandemic, reductions in homicide were one of the primary reasons why the B-W longevity gap among men shrank [18]. In 2020, however, the male B-W disparity in homicides reached its highest level since 1994 (see S1 Fig), wiping out a substantial portion of these gains.

The impact of homicide was even more pronounced in 2021. Relative to 2019, differential exposure to homicide mortality among White and Black men accounted for 26% of the increase in racial inequality in life expectancy, the second leading factor behind only COVID-19. Interpreted substantively, homicides alone accounted for a quarter of a year (0.25 years) in the change in the B-W gap in life expectancy between 2019 and 2021, an appreciably greater impact than the combined influence of suicides, other external causes, chronic lower respiratory diseases, heart diseases, Alzheimer's disease, diabetes, influenza and pneumonia, and other infectious diseases.

We now turn to the mortality shifts that occurred during the pandemic between 2020 and 2021. Following the initial devastation of the COVID-19 pandemic in 2020, the B-W life expectancy gap among men lessened in 2021, from 7.2 years to 6.0 years. As illustrated in Fig 1, deaths from COVID-19 were the largest factor by far driving this *decreasing* disparity, as White men began to succumb to COVID-19 at similar rates as Black men. On the other side of the mortality ledger, however, accidental poisoning (i.e., overdose deaths) was the leading cause of death that *increased* B-W disparities in life expectancy among men. In other words, inequality in B-W life expectancy would have decreased more in 2021 were it not for the pronounced increase in overdose mortality among Black men in 2021 relative to White men. Homicides played a relatively small role in the Black-White mortality shift between 2020 and 2021.

But the salience of homicide becomes evident when we decompose the contributions of specific causes of death to Black-White inequality in life expectancy in 2019, 2020 and 2021 separately. As shown in Fig 2, even during the peak of COVID-19 in 2020 when racial disparities in COVID-19 deaths were most acute, homicides contributed more to B-W inequality in life expectancy among men than every other cause of death. In 2021, homicide was again the leading contributor to Black-White inequality in life expectancy for men, accounting for more than twice as much of the gap as COVID-19 deaths in that year.

## Lifespan variability

As shown in Fig 3, lifespan variability is greater among Black men than White men. In each year, including the pandemic years, homicide was the primary reason why the lifespans of Black men were less certain than those of White men. For example, homicide accounted for more than half (55%) of the B-W gap in lifespan variability in 2020, more than three times the impact of COVID-19. In 2021, the impact of homicide on the B-W gap in lifespan variability was nine times greater than deaths from COVID-19 (compare 54% to 6%).

In line with previous research, we find that B-W lifespan variability decreased during the pandemic [2], driven by slight increases in variability among White men in 2020 and 2021 combined with decreased variability over this period for Black men. We examine these

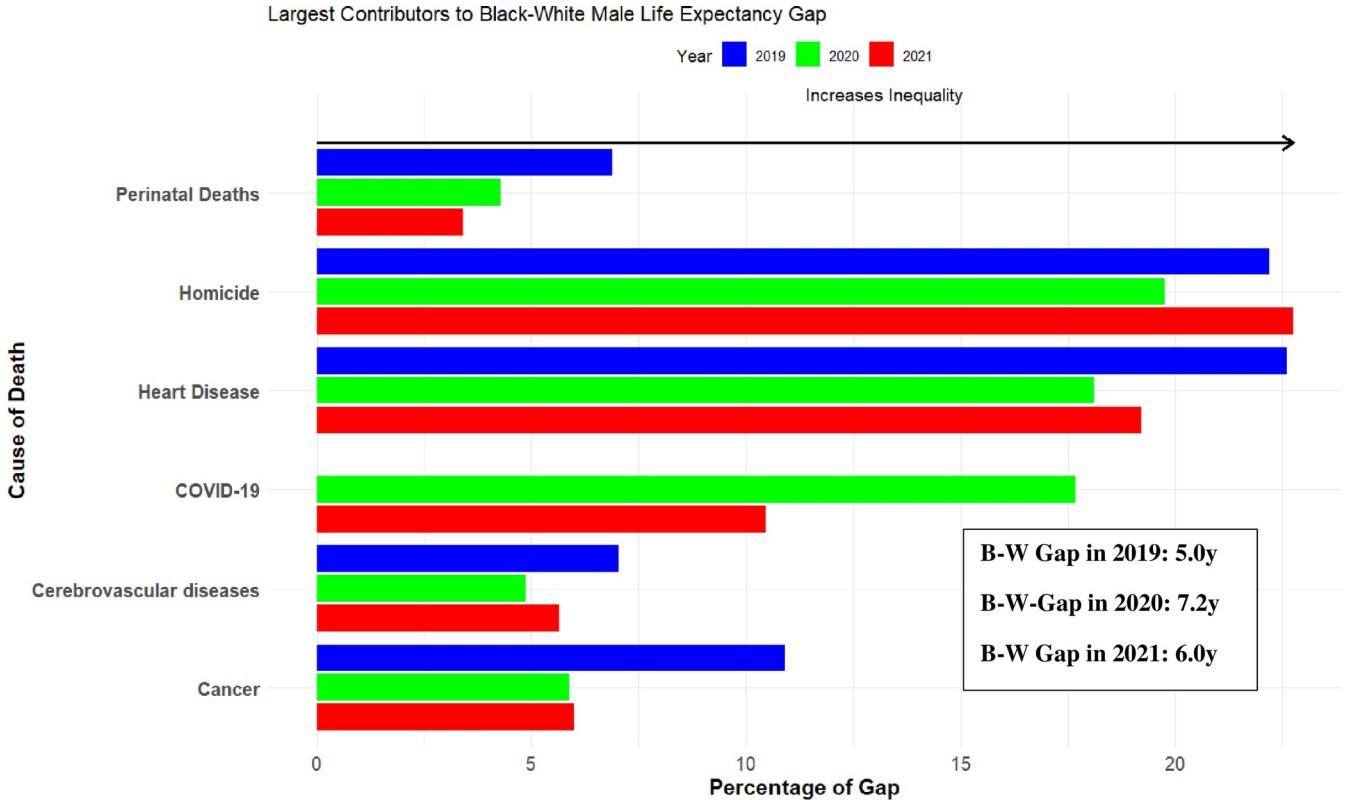

**Fig 2. Cause-specific components of the Black-White life expectancy in the United States from 2019–2021 among Males.** *Notes*: For White men, life expectancy in 2019, 2020, and 2021 was 76.4, 74.9, and 74.2 years, respectively. For Black men, life expectancy was 71.4, 67.7, and 68.1 years in 2019, 2020, and 2021.

bifurcated trends separately in Fig 4 where we look at within-race changes during the pandemic. For Black men (Panel A), most causes of death worked to decrease the amount of lifespan variability in the first year of the pandemic, save for three notable exceptions: COVID-19, homicide, and accidental poisonings (traffic accidents and diabetes also increased Black male lifespan variability but only slightly). These patterns are even more pronounced when we look at changes between 2019 and 2021 in the variance at age of death for Black men. Hence, there would have been far more certainty in the age of death among Black men were it not for the marked increases in deaths from COVID-19, homicide, and accidental poisoning.

For White men (Panel B), we see a similar pattern where most causes of death decreased the degree of lifespan variability between 2019 and 2020. However, unlike for Black men, the substantial uptick in deaths from COVID-19, overdoses, and to a lesser extent homicide, were enough to offset these decreases for White men. We observe the same general patterning in the 2019–2021 comparison among White men. Taken together, for White and especially Black men, homicide was an important contributor to lifespan variability during the COVID-19 pandemic.

## Discussion

Black-White disparities in homicide have existed for decades. But during the COVID-19 pandemic, these disparities did not just persist, they grew. Our analysis demonstrates that these increases in homicide were highly consequential for racial disparities in longevity. Simply put,

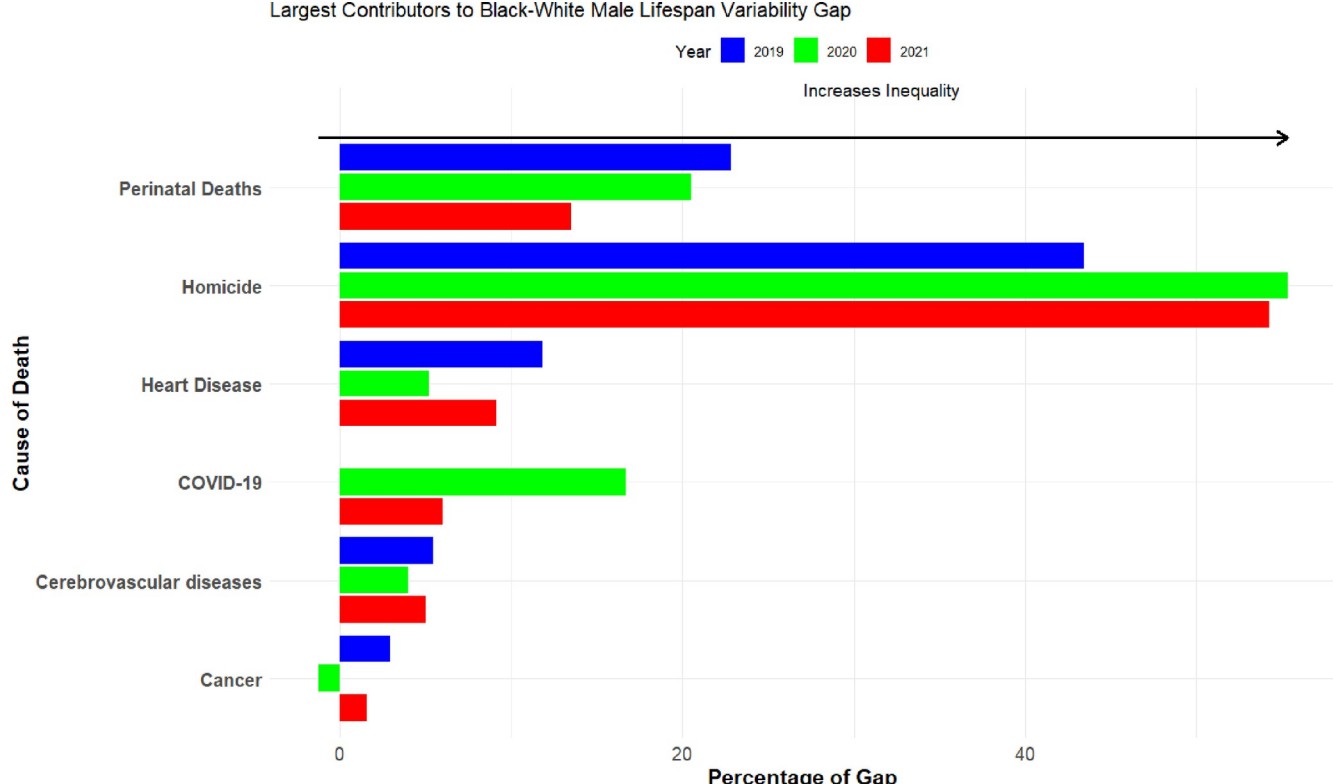

**Fig 3. Cause-specific components of the Black-White lifespan variability gap in the United States in 2019, 2020 and 2021 among Males.** *Notes*: For White men, the variance in life expectancy in 2019, 2020, and 2021 was 298.2, 302.0, and 305.7 squared-years, respectively. For Black men, variance in life expectancy was 397.0, 390.2, and 394.1 squared-years in 2019, 2020, and 2021.

homicide is one of the primary reasons why lifespans became shorter and were more variable for Black men than White men in recent years. In both 2020 and 2021, we show that homicide was the leading contributor to Black-White inequality in life expectancy and lifespan variability among men, accounting for far more of the B-W gaps in longevity and variability than deaths from COVID-19. Relative to pre-pandemic disparities, homicide was the second leading factor that drove increases in B-W inequality in longevity between 2019 and 2021.

Homicide was also among the leading factors working to increase lifespan variability among Black men. The costs of uncertain lifespans are high. If unpredictable lives result in further criminal activity [11] and less investment in long-term goals such as school, legitimate work, skill development, and retirement [10], then the social and economic costs of the substantial increase in homicide mortality among Black men in recent years may far outlast the COVID-19 pandemic.

By revealing previously overlooked sources of mortality inequality in recent years, our study identifies homicide as one of the most promising areas for policy interventions aimed at reversing recent racial disparities in longevity. This is for two reasons. First, even with the recent increases, homicides still only represent a small proportion of overall deaths in the US and less than 3% more Blacks than Whites die of homicide. Hence, our findings imply that Black-White inequality in life expectancy and lifespan variability could be narrowed substantially by eliminating this comparatively small difference. Second, there is precedent for sharp declines in homicide. Between 1991 and 2014, the homicide rate was cut by more than half, representing a "public health breakthrough for African American males, and adding 0.80 years

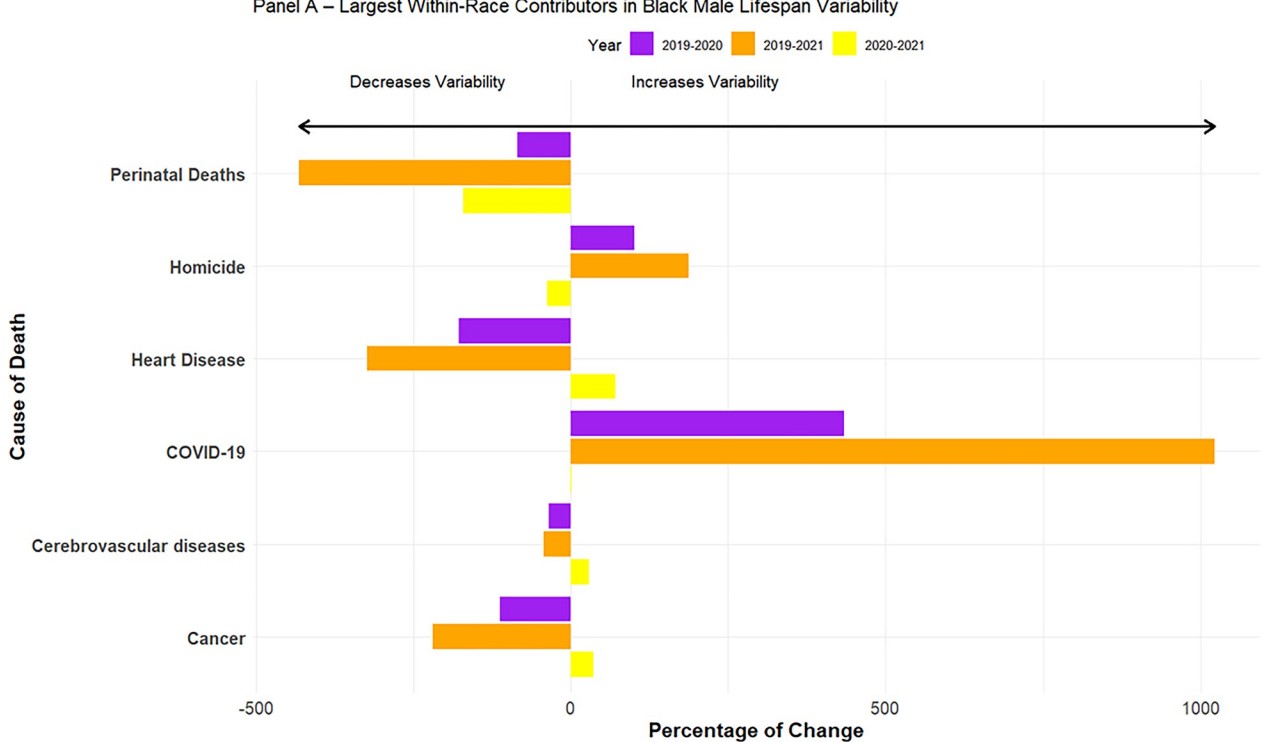

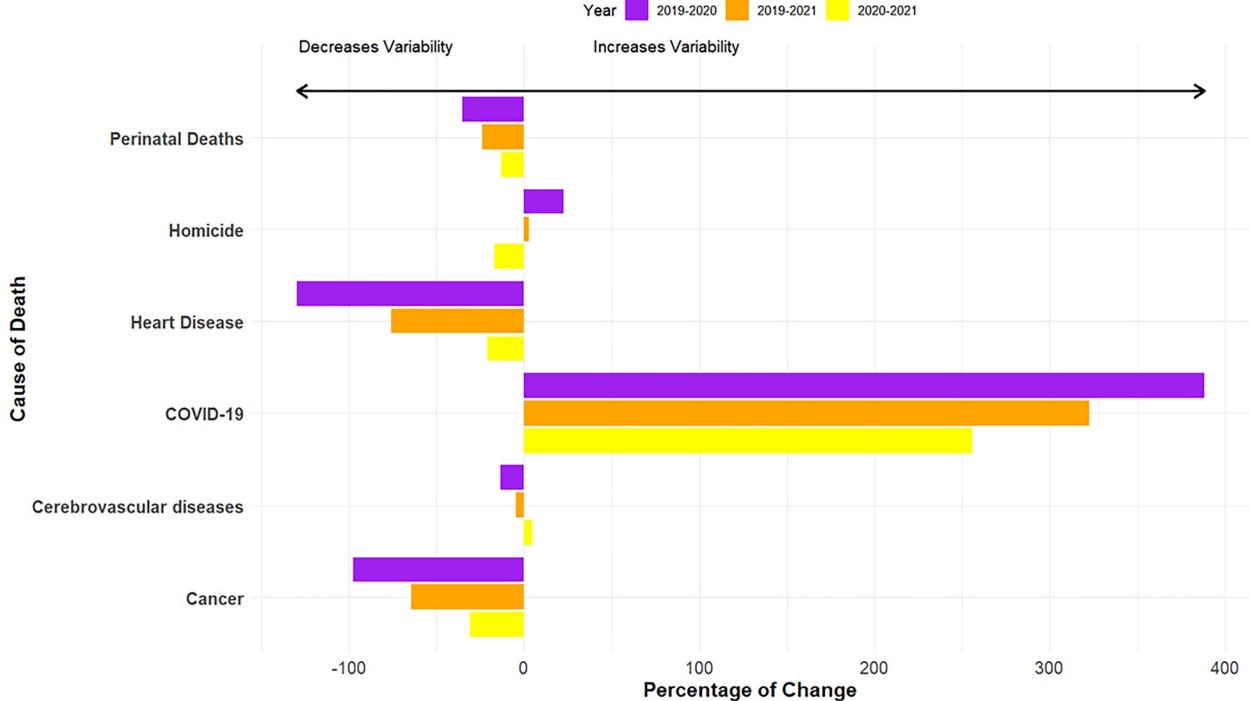

**Fig 4. Cause-specific components of the *changes* in Black Male (Panel A) and White Male (Panel B) lifespan variability in the United States from 2019–2021.** *Notes*: For Black men, variance in life expectancy was 397.0, 390.2, and 394.1 squared-years in 2019, 2020, and 2021.For White men, the variance in life expectancy in 2019, 2020, and 2021 was 298.2, 302.0, and 305.7 squared-years, respectively.

to life expectancy at birth. . ." (18: 658). The causes of the "Great American Crime Decline" [19] have been subject to a tremendous amount of research [20,21], and while there is disagreement about the relative weight of various factors, this body of work still provides evidence for several encouraging policy interventions for curbing violence, including investments in community nonprofit organizations and policing [22]. One thing is clear: reductions in homicide will require conscious effort and the policies designed to lower violence are likely to be markedly different from those aimed at other sources of racial inequality in life expectancy (e.g., heart diseases).

Although the CDC and Census data have been widely used to study longevity, readers should consider limitations in mortality and population data when interpreting our results. The underlying cause of death could be miscoded, and prior research suggests that such errors correlate with race [23]. Along similar lines, mortality estimates could be biased by population undercounts, age misreporting, and racial misclassifications. For our longitudinal analyses, such concerns are likely minimal because any non-random recording errors would have to systematically change in a short period of time, and there is scant evidence this occurred. However, data errors could play a larger role in our cross-sectional decompositions.

## Conclusion

Differences in longevity are a primary source of social inequality more broadly [24], and our study reveals that differential exposure to homicide played a central role in increasing B-W inequality in life expectancy during the COVID-19 pandemic among men. These results are particularly sobering in light of the most recent trends in COVID-19 and homicide deaths. COVID-19 deaths declined by 47% in 2022 [25], while homicide deaths were largely stable [26]. The homicide data from 2023 looks more encouraging, with notable decreases in many major cities throughout the US compared to the same time in 2022. However, even with record double-digit declines in 2023, there were still substantially more homicides in the US compared to 2019 [27]. Thus, given our results, the impact of homicides on Black-White inequality in life expectancy and variability may be even more pronounced in 2022 and 2023. Therefore, reductions in violence should be at the fore of any public health discussion aimed at promoting health equity.

## Supporting information

**S1 Data. Data and materials availability.**
(DOCX)

**S1 Fig. Differences in homicide death rates between Black and White men, 1990–2021.**
(DOCX)

**S1 Table. Black-White male cause decomposition by year.**
(DOCX)

**S2 Table. Variance decomposition by race and year for men.**
(DOCX)

**S3 Table. Cause-grouping and corresponding ICD-10 codes.**
(DOCX)

## Acknowledgments

We thank Marcus Felson, Jenna Nobles, and the anonymous reviewers for their helpful comments.

## Author Contributions

**Conceptualization:** Michael T. Light.

**Data curation:** Karl Vachuska.

**Formal analysis:** Karl Vachuska.

**Methodology:** Michael T. Light, Karl Vachuska.

**Supervision:** Michael T. Light.

**Visualization:** Karl Vachuska.

**Writing – original draft:** Michael T. Light.

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
