## [Decision Letter · Decision Letter 0]

5 Apr 2024

PONE-D-23-42765Increased homicide played a key role in driving black-white disparities in life expectancy during the COVID-19 pandemicPLOS ONE

Dear Dr. Light,

Thank you for submitting your manuscript to PLOS ONE. After careful consideration, we feel that it has merit but does not fully meet PLOS ONE’s publication criteria as it currently stands. Therefore, we invite you to submit a revised version of the manuscript that addresses the points raised during the review process.

**ACADEMIC EDITOR: **I regret to inform you that we are unable to accept your manuscript for publication in its current form.

The reviewers have provided valuable feedback on your manuscript, with two out of three reviews being positive. 

However, the negative review raises significant concerns about the research methodology and the life expectancy estimates. 

Therefore, we recommend that you thoroughly address all the comments provided by the reviewers and revise your manuscript accordingly.

It is essential that you address all these concerns comprehensively in your revised manuscript to ensure its suitability for publication.

We understand that revising your manuscript may require substantial effort, but we believe that addressing the reviewers' comments will significantly improve the quality and impact of your research. We encourage you to carefully consider all the feedback provided and to make the necessary revisions accordingly. 

We look forward to receiving your revised manuscript.

Kind regards,

Claudio Alberto Dávila-Cervantes, Ph.D.

Academic Editor

PLOS ONE

2. Please note that PLOS ONE has specific guidelines on code sharing for submissions in which author-generated code underpins the findings in the manuscript. In these cases, all author-generated code must be made available without restrictions upon publication of the work. Please review our guidelines at https://journals.plos.org/plosone/s/materials-and-software-sharing#loc-sharing-code and ensure that your code is shared in a way that follows best practice and facilitates reproducibility and reuse."

 [This research is supported by the Romnes Faculty Fellowship provided by the University of Wisconsin-Madison Office of the Vice Chancellor for Research and Graduate Education with funding from the Wisconsin Alumni Research Foundation.].  

5. We notice that your supplementary figures and tables are included in the manuscript file. Please remove them and upload them with the file type 'Supporting Information'. Please ensure that each Supporting Information file has a legend listed in the manuscript after the references list.

6. Please remove your figures from within your manuscript file, leaving only the individual TIFF/EPS image files, uploaded separately. These will be automatically included in the reviewers’ PDF.

Reviewers' comments:

Reviewer's Responses to Questions

**Comments to the Author**

1. Is the manuscript technically sound, and do the data support the conclusions?

Reviewer #1: Yes

Reviewer #2: No

Reviewer #3: Yes

2. Has the statistical analysis been performed appropriately and rigorously? 

Reviewer #1: Yes

Reviewer #2: No

Reviewer #3: Yes

3. Have the authors made all data underlying the findings in their manuscript fully available?

Reviewer #1: Yes

Reviewer #2: No

Reviewer #3: Yes

4. Is the manuscript presented in an intelligible fashion and written in standard English?

Reviewer #1: Yes

Reviewer #2: Yes

Reviewer #3: Yes

5. Review Comments to the Author

Reviewer #1: It was a pleasure to read this well-written and carefully executed study of Black-White longevity disparities during the pandemic.

My only questions to the authors are: (1) Was the initial 5-year age category separated into 0-1 and 1-4 age intervals, so as to separate perinatal/infant from early childhood mortality?, and (2) What methodology was used to estimate number of years lived among those who died within each of the age categories (i.e., nax)?

Those minor issues/questions aside, I think this is a terrific manuscript. The topic is timely and important, the demographic methods are sound, the data while imperfect are high quality (particularly relative to some other sources of observational data in social science and epidemiology), and the overarching conclusion – that more attention needs to be paid to homicide as a driver or Black-White disparities in mortality – is well supported by the findings. In addition, the focus on mortality variability and the effects this could have on discounting the future among Black youth is interesting and will make an important contribution to the literature.

Reviewer #2: The manuscript “Increased Homicide Played a Key Role in Driving Black-White Disparities in Life Expectancy during the COVID-19 Pandemic” estimates non-Hispanic Black and non-Hispanic white life expectancy and lifespan variability in the United States for the years 2019, 2020, and 2021. The authors then decompose the changes into cause-specific contributions and argue that homicide deaths in 2021 was a primary contributor to inequalities between Black and white men between 2019 and 2021.

This is the first time I am reviewing this manuscript. While the topic is likely to of broad interest to both scholars and policymakers, I have some major concerns with the methods and caution that the paper is not ready for acceptance. Below I organize my thoughts around (1) major concerns, and (2) smaller points. I hope that the authors find them useful as they continue to rework this manuscript.

Major

The life expectancy estimates are quite different from what I have seen in other outlets, including the final life expectancy estimates from the NVSS. For example, the 5.5 year life expectancy decline (77.7 to 72.2) that the authors observe for Black males between 2019 and 2020 is very different than estimates reported elsewhere: (a) in Aburto et al: 71.4 (2019) to 67.8 (2020) – loss of 3.6 years, (b) in Masters et al: 74.78 (2019) to 71.55 (2020) – loss of 3.23 years, and (c) in US Life Tables: 74.8 (2019) to 71.5 (2020) – loss of 3.3 years. The initial levels and the drop are quite different between this manuscript and other work.

I wish I could have looked through the authors’ code to understand where this discrepancy was coming from, but unfortunately it was not shared with reviewers. Because I do not trust even the baseline estimates of life expectancy that are reported here, I have a hard time trusting the remainder of the authors’ results.

Related, is it actually standard to assume an age of death of 90 for all those over 85 (as the authors say in the methods)? In my experience, it is more common to use penalized composite link methodology (PCLM) to smooth mortality and population counts out to some determined age (usually to 110). Stopping counts at 85+ (or assuming a death of 90 for all over 85) biases mortality estimates. Perhaps this is where the strange life expectancy estimates are coming from?

Smaller Points

• From the introduction: “Not only did the disease burden of the COVID-19 virus fall disproportionately on Black Americans, but the cascading disruptions during the pandemic also took a heavy toll on the Black community, perhaps none more so than homicide.”

o This feels a bit suggestive. It statement implies that the rise in homicide during the pandemic years outweighed the burden of other causes of death (e.g., CVD or Drug-related deaths) and I’m not sure that is the case.

• I wonder about the language of “murder” vs. “homicide.” There are some meaningful differences between the two, and I would encourage the authors to be mindful of using the correct terminology throughout.

• A very minor point: “For non-Hispanic White men, the increase was only 22%” (Page 3, line 43) – this is still a big increase. I recommend removing the word “only.”

References

Aburto, J. M., Tilstra, A. M., Floridi, G., & Dowd, J. B. (2022). Significant impacts of the COVID-19 pandemic on race/ethnic differences in US mortality. Proceedings of the National Academy of Sciences, 119(35), e2205813119.

Masters, R. K., Aron, L. Y., & Woolf, S. H. (2024). Life Expectancy Changes During the COVID-19 Pandemic, 2019–2021: Highly Racialized Deaths in Young and Middle Adulthood in the United States as Compared With Other High-Income Countries. American Journal of Epidemiology, 193(1), 26-35.

US Life Table: https://dx.doi.org/10.15620/cdc:118055

Reviewer #3: “Increased homicide played a key role in driving black-white disparities in life expectancy during the COVID-19 pandemic”

By Michael T. Light and Karl Vachuska

The manuscript by Light and Vachuska investigates an important topic- the effect that homicide mortality had on Black:White inequities in life expectancy during the COVID-19 pandemic. The authors investigated both changes in average life expectancy and lifespan variability. The authors conclude that homicide contributed to why lifespans became shorter and more variable for Blacks compared to Whites in the period 2019-2021.

I thought that this manuscript was quite well-written. The methodology consisted of life tables and multiple decrement processes has been used previously (Aburto et al 2022) and is widely concerned adequate and appropriate for this area of investigation.

Some comments:

1. Early in the manuscript (line 86) the authors describe categorizing causes of death into groups based on common etiologies (external causes, chronic disease, communicable diseases and a residual category). But it seems that nothing further was done with this categorization. When the authors describe contributions of specific causes, they do so by those causes (heart disease, traffic accidents, etc.). Because the authors don’t elaborate further on these categorizations, reference to them should be removed from the manuscript.

2. A comment on the tables and figures: this may be something that the authors cannot do anything about- but the current format of the tables and figures is incredibly confusing and requires so much more time and effort to comprehend than is reasonable. For example, Panel A. 2020 on page 23 does not have labels to indicate whether this was for males, females or both, and whether this figure is for average life expectancy or lifespan variability. I am not sure what the solution is- the authors’ original intent (to categorize deaths into four groups of causes) might have been helpful here- as the inclusion of a bunch of other causes, that are not further explained or discussed (i.e. cerebrovascular diseases and accidental poisoning) take up valuable real estate that could be better spent illustrating the findings of the paper).

There is uneven treatment of the tables and figures more generally. The next page (24) includes a title (Figure 2. Cause-specific components of the Black-White life expectancy gap in the United States in 2020 and 2021 among Males). There is also a note indicating what the ALE was for black and white males in 2020 and 2021-but nowhere is the actual number of years shown anywhere. One possible solution would be to refer to actual number of years of the life expectancy gap. Possibly both could be included- as it is now, it is not clear what I am looking at on page 24- I think it is the comparison for males in 2021. But, why information about 2020 was included in the Figure caption is not known. Basically, the reader is guessing what is happening in each of the tables. I think the authors could improve the figures.

3. I was rather confused about the scope of this manuscript- while I initially thought that it focused on males and females together, and the population more broadly, the authors relegated results regarding women to the SI appendix. Also, in the supplement, there was a line graph for men, but not for women. The overall result seems lopsided and awkward. Moreover, I think that the results for men were rather stark and I would support narrowing the scope of this manuscript to include only male mortality. Certainly, interventions aimed at reducing homicide mortality for men and women may look different- particularly since most female homicide mortality is related to intimate partner violence.

4. I think that the authors statement “By revealing previously overlooked sources of mortality inequality in recent years, our study recalibrates our understanding of the drivers of Black-White disparities in life expectancy during the pandemic” is a bit over stated. Sub-national studies in mortality have shown changes in homicide to be a major contributor to dynamic inequities in average life expectancy, particularly among males see (Friedman et al 2017; Fenelon and Boudroux 2019; Bishop-Royse 2023).

6. PLOS authors have the option to publish the peer review history of their article (what does this mean?). If published, this will include your full peer review and any attached files.

Reviewer #1: No

Reviewer #2: No

Reviewer #3: No

---

## [Author Response · Author response to Decision Letter 0]

7 Jun 2024

Please see the detailed response memo submitted.

---

## [Decision Letter · Decision Letter 1]

17 Jul 2024

Increased homicide played a key role in driving black-white disparities in life expectancy among men during the COVID-19 pandemic

PONE-D-23-42765R1

Dear Dr. Light,

We’re pleased to inform you that your manuscript has been judged scientifically suitable for publication and will be formally accepted for publication once it meets all outstanding technical requirements.

Kind regards,

Claudio Alberto Dávila-Cervantes, Ph.D.

Academic Editor

PLOS ONE

Reviewers' comments:

Reviewer's Responses to Questions

**Comments to the Author**

1. If the authors have adequately addressed your comments raised in a previous round of review and you feel that this manuscript is now acceptable for publication, you may indicate that here to bypass the “Comments to the Author” section, enter your conflict of interest statement in the “Confidential to Editor” section, and submit your "Accept" recommendation.

Reviewer #3: All comments have been addressed

2. Is the manuscript technically sound, and do the data support the conclusions?

Reviewer #3: Yes

3. Has the statistical analysis been performed appropriately and rigorously? 

Reviewer #3: Yes

4. Have the authors made all data underlying the findings in their manuscript fully available?

Reviewer #3: Yes

5. Is the manuscript presented in an intelligible fashion and written in standard English?

Reviewer #3: Yes

6. Review Comments to the Author

Reviewer #3: Excellent article. I appreciate the care that the reviewers took in addressing my concerns (as well as those of the other reviewers). I think that the article was strengthened by narrowing the focus on male mortality. I believe that this manuscript is greatly improved.

7. PLOS authors have the option to publish the peer review history of their article (what does this mean?). If published, this will include your full peer review and any attached files.

Reviewer #3: No

---

## [Editor Report · Acceptance letter]

24 Jul 2024

PONE-D-23-42765R1 

PLOS ONE

Dear Dr. Light, 

I'm pleased to inform you that your manuscript has been deemed suitable for publication in PLOS ONE. Congratulations! Your manuscript is now being handed over to our production team.

Kind regards, 

on behalf of

Mr. Claudio Alberto Dávila-Cervantes 

Academic Editor

PLOS ONE